# Learning Entailment-Based Sentence Embeddings from Natural Language Inference

## Abstract

Large datasets on natural language inference are a potentially valuable resource for inducing semantic representations of natural language sentences. But in many such models the embeddings computed by the sentence encoder goes through an MLP-based interaction layer before predicting its label, and thus some of the information about textual entailment is encoded in the interpretation of sentence embeddings given by this parameterised MLP. In this work we propose a simple interaction layer based on predefined entailment and contradiction scores applied directly to the sentence embeddings. This parameter-free interaction model achieves results on natural language inference competitive with MLP-based models, demonstrating that the trained sentence embeddings directly represent the information needed for textual entailment. The inductive bias of this model leads to better generalisation to other natural language inference datasets, and the resulting sentence embeddings are better for other semantic tasks.

## 1 Introduction

After the enormous success of learning representations of words, learning representations of sentences has become one of the most important challenges in natural language understanding. Many models have been proposed for embedding sentences in a vector space, but the methods for interpreting those representations are dominated by two approaches: using the dot product (or cosine) to measure the similarity between two vectors, or training a parameterised function to do the interpretation for specific tasks (e.g. (Conneau et al., 2017)). In this paper we consider a different form of interpretation, based neither on similarity nor on a parameterised function. We investigate embedding sentences into vector representations of information inclusion and contradiction.

### 1.1 Motivation

Information inclusion, called entailment, is a fundamental concept in theories of the semantics of natural language. This motivated the proposal of recognising entailment in text as a generic task which captures a very wide range of issues in natural language understanding (Dagan et al., 2005), which has been demonstrated by the conversion of a large number of semantic datasets into a single textual entailment task (Poliak et al., 2018a). The fundamental nature of this task has motivated the development of many datasets for textual entailment (Poliak et al., 2018a). The more recent textual entailment datasets have extended this task to include annotations for contradiction, which are often referred to as natural language inference (NLI) datasets.

Modelling these semantic relationships between sentences is not only a crucial problem in natural language processing, but it also has a wide variety of applications, such as question answering (Harabagiu & Hickl, 2006), producing abstract summaries (Lacatusu et al., 2006; Yan et al., 2011a;b), and machine translation evaluation (Padó et al., 2009). It is possible to transfer the predictions of an NLI model directly to the task, but it has been shown that NLI models trained on large NLI benchmarks such as SNLI (Bowman et al., 2015) and MNLI (Williams et al., 2018a) do not transfer well to other NLI benchmarks (Talman & Chatzikyriakidis, 2019). Despite recent efforts to improve generalisation (Belinkov et al., 2019b), this remains an open question. A more flexible and powerful approach is to transfer the knowledge learned about the NLI task to other tasks with representation learning. For example, Conneau et al. (2017) train sentence embeddings on the NLI

task and demonstrate their usefulness on other tasks. But there is still no clear interpretation of these embeddings in terms of entailment, other than via a trained parameterised function.

In this work we propose a novel approach to learning sentence embeddings from NLI data, where the sentence embeddings have a direct interpretation in terms of entailment and contradiction. Analogously to the use of the dot product or cosine to define similarity over a vector space, we define entailment and contradiction as two parameter-free operators over pairs of vectors. The entailment operator provides a score of how well one vector includes all the information in the other vector. The contradiction operator provides a score of to what extent two vectors contradict each other. We use these operators to train sentence encoders which embed sentences into a vector space which has these interpretations, using data for the Natural Language Inference task.

## 1.2 OVERVIEW

Natural language inference involves predicting whether a *premise* entails the given *hypothesis*, whether they contradict each other, or neither (Bowman et al., 2015). Due to the availability of large-scale datasets for natural language inference (Bowman et al., 2015; Williams et al., 2018b), neural network models have become popular for textual entailment. We can characterise these models in terms of three stages. First, the encoding stage computes $d$-dimensional embeddings of the premise and of the hypothesis. Then the *interaction layer* models the interactions between the two sentences. Finally a classifier layer makes the prediction of entailment, contradiction or neutral. While most of the literature has been concerned with improving the encoding stage, in this paper we focus on the interaction layer.

For the interaction layer, passing the sentence embeddings plus their *heuristic matching features* (Mou et al., 2016) through a multi-layer perceptron (MLP) has become the standard practice for textual entailment models (Conneau et al., 2017; Bowman et al., 2016; Kim et al., 2019; Pan et al., 2018; Yoon et al., 2018; Kiela et al., 2018; Conneau et al., 2018b). The heuristic matching features are component-wise measures of similarity between premise and hypothesis, most commonly implemented as the absolute difference and the component-wise multiplication. However, textual entailment is not only about similarity, since it reflects an information inclusion relationship. Moreover, most similarity features are symmetric, whereas entailment is an intrinsically asymmetric relation. It is only after the MLP that the sentence embeddings and similarity-based features get transformed into a representation which can be directly mapped to entailment, contradiction and neutral. Often the MLP plays a significant role in the model's architecture, for example requiring 800k of the total 2.1m model parameters in the state-of-the-art Dynamic Self-Attention model (Yoon et al., 2018).

In this paper, we propose a parameter-free interaction layer consisting of only a few scores, including entailment and contradiction, defined in terms of the entailment-vectors framework of Henderson & Popa (2016), which are passed to a log-linear classification layer. This forces the sentence embeddings to have a direct interpretation in terms of entailment and contradiction, and imposes an inductive bias towards learning entailment and contradiction.

Despite drastically reducing the number of trained parameters, this simple interaction layer is empirically competitive with MLP-based models. Because it is parameter-free, all of the information learned about entailment and contradiction must be encoded directly in the sentence embeddings, rather than indirectly via its interpretation by the trained MLP. Thus information about entailment or contradiction can be transferred to other tasks simply by transferring the sentence embeddings. An ablation analysis indicates that the entailment and contradiction scores are indeed the main factors in the model's success. When we test our trained models on related datasets that have not been used for training, we find that the inductive bias of our proposed model results in better transfer performance. When we test our induced sentence embeddings on other tasks, we find better transfer performance on semantic tasks.

In summary, we propose a novel, theoretically grounded, parameter-free model of the interactions between premise and hypothesis for entailment and contradiction. Evaluation on benchmark natural language inference datasets show this model is competitive with modelling interaction with an MLP. The resulting sentence embeddings have a direct entailment-based interpretation, and the model's inductive bias results in better transfer to different NLI datasets and different semantic tasks.

## 2 MODELLING ENTAILMENT AND CONTRADICTION RELATIONS

The task of textual entailment takes premise and hypothesis sentences and classifies their relationship into `entailment`, `contradiction`, or `neutral`. To train our sentence embeddings, we adopt the widely used approach of first encoding each sentence in an embedding, then modelling the interactions between the premise and hypothesis, followed by a softmax classifier which estimates the probability of each label (Conneau et al., 2018b).

### 2.1 ENCODING STAGE

The encoding stage embeds the premise $s1$ and hypothesis $s2$ into two $d$-dimensional vector representations $\boldsymbol{p}, \boldsymbol{h} = \text{enc}(s1, s2)$, where enc is a sentence encoder, and $\boldsymbol{p}$ and $\boldsymbol{h}$ are the premise and hypothesis embeddings respectively. There are two types of encoders; intra-sentence encoders only look at each sentence individually, while inter-sentence encoders consider both sentences. In our first set of experiments, we consider models with both single-sentence and inter-sentence encoders, specifically the two baselines from the GLUE benchmark (Wang et al., 2019). The single-sentence encoder is a bidirectional LSTM (Bi-LSTM) with max-pooling, which Conneau et al. (2017) found to work best among a diverse set of encoders. The inter-sentence encoder adds attention over the other sentence (Wang et al., 2019). In our second set of experiments, we consider models with the single-sentence encoder of (Reimers & Gurevych, 2019), based on a pre-trained BERT model Devlin et al. (2019).

### 2.2 MLP INTERACTION STAGE

Most previous neural network approaches to NLI model the interaction between sentences with a non-linear MLP. The input to this MLP includes the embeddings of the hypothesis $\boldsymbol{h}$ and premise $\boldsymbol{p}$ obtained from the encoding stage. Mou et al. (2016) proposed adding *heuristic matching features* in order to capture the similarity of the two sentences, which have since been widely adopted (Conneau et al., 2017; Chen et al., 2017; Bowman et al., 2016; Kim et al., 2019; Pan et al., 2018; Yoon et al., 2018; Kiela et al., 2018). These features are most commonly the element-wise product and absolute difference between the two sentence embeddings, giving the input vector $\boldsymbol{m}$:

$$\boldsymbol{m} = [\boldsymbol{p}; \boldsymbol{h}; |\boldsymbol{p} - \boldsymbol{h}|; \boldsymbol{p} \odot \boldsymbol{h}] \qquad (1)$$

The MLP then computes a hidden layer, from which the softmax classifier predicts the labels.

$$\text{class} = \text{softmax}(\boldsymbol{W}_c \tanh(\boldsymbol{W}_e \boldsymbol{m} + \boldsymbol{b}_e) + \boldsymbol{b}_c) \qquad (2)$$

where $\boldsymbol{W}_c \in \mathbb{R}^{3 \times n}, \boldsymbol{b}_c \in \mathbb{R}^3, \boldsymbol{W}_e \in \mathbb{R}^{n \times 4d}, \boldsymbol{b}_e \in \mathbb{R}^n$, and $n$ is the size of the hidden layer. Because of the high dimensionality $d$ of the sentence embeddings, the number of parameters in this interaction layer ($\boldsymbol{W}_e$) can be large. Following Wang et al. (2019), we set $n = 512$.

### 2.3 THE PROPOSED INTERACTION STAGE

We propose an alternative interaction stage with no parameters. The input is the two sentence embeddings, and the output is just 5 scores, which are then input to the softmax classifier. These predefined scores provide a strong inductive bias to the model, and force the sentence embeddings to have the interpretation required by these scores.

**Normalisation** In the 5 scores, the sentence embeddings are first passed through a sigmoid function, and we want them to use the full output range of this function. We therefore apply layer normalisation to the sentence embeddings to make their mean zero and their standard deviation $s$, where $s$ is a hyperparameter set to 6 in all experiments. We use $\boldsymbol{y}, \boldsymbol{x}$ to refer to the embeddings $\boldsymbol{p}, \boldsymbol{h}$ respectively after applying normalisation.

**Entailment Vectors Framework** Henderson & Popa (2016) propose a vector-space framework for entailment which explicitly models the information known about a word by interpreting each dimension of a word embedding as the probability that a certain feature is known to be present. So zero is interpreted as unknown, and not as necessarily false. Entailment is when all the features that are known about the entailed vector are also known about the entailing vector. They propose

different operators for computing the probability of this entailment. Previous work only applied this framework to *lexical* entailment (Henderson & Popa, 2016), but we apply it to *textual* entailment.

**Modelling Contradictory Information**  The original entailment vectors framework only models binary entailment. We extend this representation to also model contradiction. We split entailment vectors into two blocks, one $\boldsymbol{x}_{1:\frac{d}{2}}$ for modelling features $f$ and another $\boldsymbol{x}_{\frac{d}{2}:d}$ for their negation $\neg f$ (as suggested by Henderson & Popa (2016) for a different purpose). So if $\boldsymbol{x}_k$, $k \in \{1, \ldots, \frac{d}{2}\}$, represents that something is known to be true, then $\boldsymbol{x}_{k+\frac{d}{2}}$ represents that it is known to be false, and thus having both these features active is a contradiction.

**Entailment Score:**  We compute the entailment *score* between two sentences using the factorised *entailment operator* $(\boldsymbol{y} \dot{\Rightarrow} \boldsymbol{x})$ proposed by Henderson & Popa (2016), which measures the probability of entailment in the k-th dimension as $S_k(\text{entail}|\boldsymbol{x}, \boldsymbol{y}) = 1 - \sigma(-\boldsymbol{y}_k)\sigma(\boldsymbol{x}_k)$, and which requires entailment to be probable in every dimension, giving us:

$$S(\text{entail}|\boldsymbol{x}, \boldsymbol{y}) = (\prod_{k=1}^{d} 1 - \sigma(-\boldsymbol{y}_k)\sigma(\boldsymbol{x}_k))^{\frac{1}{d}} \tag{3}$$

**Contradiction Score:**  We consider two sentences to contradict each other if their respective embeddings have contradictory features in at least one dimension. Our contradiction score for the k-th dimension $\in \{1, \ldots, \frac{d}{2}\}$ is thus:

$$S_k(\text{contradict}|\boldsymbol{x}, \boldsymbol{y}) = \sigma(\boldsymbol{x}_k)\sigma(\boldsymbol{y}_{k+\frac{d}{2}}) + \sigma(\boldsymbol{x}_{k+\frac{d}{2}})\sigma(\boldsymbol{y}_k) - \sigma(\boldsymbol{x}_k)\sigma(\boldsymbol{y}_{k+\frac{d}{2}})\sigma(\boldsymbol{x}_{k+\frac{d}{2}})\sigma(\boldsymbol{y}_k)$$

where the first term models that the feature is known to be true in the hypothesis and known to be false in the premise, the second term the reverse, and the third term both. The total score of contradiction is the complement of not having contradiction in any dimension:

$$S(\text{contradict}|\boldsymbol{x}, \boldsymbol{y}) = 1 - (\prod_{k=1}^{\frac{d}{2}} (1 - S_k(\text{contradict}|\boldsymbol{x}, \boldsymbol{y})))^{\frac{1}{d}}$$

**Neutral Score:**  We define a neutral score as the non-negative complement of the contradiction and entailment scores, computed as:

$$S(\text{neutral}|\boldsymbol{x}, \boldsymbol{y}) = \text{ReLU}(1 - S(\text{entail}|\boldsymbol{x}, \boldsymbol{y}) - S(\text{contradict}|\boldsymbol{x}, \boldsymbol{y})),$$

The ReLU function avoids negative scores, and its nonlinearity makes this score non-redundant in the log-linear softmax classifier.

**Similarity Scores:**  Since similarity information is known to help  textual entailment (Mou et al., 2016), we employ two scores that can be viewed as condensed versions of the heuristic matching features,  namely averages of element-wise multiplication and absolute difference, respectively:

$$sim_{mul}(\boldsymbol{x}, \boldsymbol{y}) = \frac{1}{d} \sum_{k=1}^{d} (\sigma(\boldsymbol{x}_k)\sigma(\boldsymbol{y}_k))$$

$$sim_{diff}(\boldsymbol{x}, \boldsymbol{y}) = \frac{1}{d} \sum_{k=1}^{d} (|\sigma(\boldsymbol{x}_k) - \sigma(\boldsymbol{y}_k)|).$$

**Initialisation**  For the success of our models, we found it critical to initialise the output layer in such a way that the scores are associated with their respective class label. To this end, we add 1 to the random initial weight connecting the entailment score to the entailment class, the contradiction score to the contradiction class, and the neutral score to the neutral class, respectively. We observed that without this initialisation the model takes several epochs to train before achieving better than random performance.

## 3 EVALUATION ON NLI DATASETS

We first evaluate our approach to learning sentence embeddings from NLI data without using any resources external to the NLI datasets themselves. We train and test our models on two popular large-scale datasets for textual entailment, SNLI (Bowman et al., 2015) and MNLI (Williams et al., 2018a). In addition to evaluating classification performance, we conduct an ablation study to determine whether our sentence embeddings and associated scores have the interpretations claimed.

### 3.1 EXPERIMENTAL SETUP

For the LSTM-based sentence encoders, we always use Bi-LSTM state vectors with 512 dimensions. With larger models (2048 dimensions) the results were only slightly better and the pattern of results did not change, while the computational cost was substantially higher. We train these models for 20 epochs, use validation accuracy for early stopping, and use mini-batch SGD with batch size of $64$. The learning rate is selected among $\{0.06, 0.08, 0.1, 0.3, 0.5\}$ based on the validation scores.

For MNLI, we tuned the models on the mismatched validation set, and report results for both mismatched and matched validation sets, only reporting test set results for the best baseline and our best models.

### 3.2 BASELINES

We compared our model with several baselines: 1) *p,h*: Using only sentence embeddings followed by an MLP; 2) *HM*: Heuristic-matching features input to an MLP (i.e. the baseline from Conneau et al. (2017)); 3) *Random*: compute five random scores followed by a linear softmax classifier. For baseline (3), we use *untrained* nonlinear projections of the embeddings $\boldsymbol{p}, \boldsymbol{h}$ of the form:

$$\boldsymbol{r} = \sigma(\boldsymbol{W}_g \sigma(\boldsymbol{W}_i[\boldsymbol{p}; \boldsymbol{h}] + \boldsymbol{b}_i) + \boldsymbol{b}_g), \tag{4}$$

where the weight matrices $\boldsymbol{W}_i \in \mathbb{R}^{d \times 2d}$, $\boldsymbol{W}_g \in \mathbb{R}^{5 \times d}$ and their corresponding biases are randomly generated using Glorot Initialisation (Glorot & Bengio, 2010). This baseline evaluates how much performance is being gained by employing our specific five scores, with their inductive bias.

Table 1: Validation and test accuracies (for MNLI, on matched/mismatched sets), and number of parameters in the encoder (*#enc*) and MLP and/or classifier (*#mlp*).

| Model | #enc | #mlp | SNLI | MNLI |
|---|---|---|---|---|
| Random | 3.3m | 18 | 79.07 | 65.91 / 65.88 |
| p,h | 3.3m | 1.3m | 78.70 | 64.70 / 65.69 |
| HM | 3.3m | 2.4m | 84.82 | 71.23 / 71.46 |
| Ours | 3.3m | 18 | 83.47 | 69.97 / 70.51 |
| HM+attn | 13.8m | 2.4m | 86.46 | 74.81 / 74.81 |
| Ours+attn | 13.8m | 18 | 86.28 | 74.21 / 74.41 |
| **Test set results** | | | | |
| HM | 3.3m | 2.4m | 84.79 | 71.22 / 70.36 |
| Ours | 3.3m | 18 | 83.09 | 69.79 / 68.96 |
| HM+attn | 13.8m | 2.4m | 85.52 | 74.37 / 73.60 |
| Ours+attn | 13.8m | 18 | 85.49 | 73.14 / 72.74 |

Table 2: Ablation results for the scores (C)ontradiction, (E)ntailment, (N)eutral, and (S)imilarity, on SNLI and MNLI (matched/mismatched) validation sets.

| Used scores | SNLI | MNLI |
|---|---|---|
| E, C, N, S | 83.47 | 69.97 / 70.51 |
| E, C, N | 83.14 | 69.19 / 69.97 |
| E, C | 78.02 | 69.49 / 69.66 |
| S | 75.48 | 63.03 / 63.31 |
| E | 78.62 | 63.57 / 63.92 |
| C | 74.7 | 58.19 / 58.96 |

### 3.3 RESULTS

Table 1 shows the results. First, note that random scores already achieve reasonable performance, at about the level of the MLP-based baseline p,h. This suggests that the encoder is so powerful that it can generate embeddings suitable for virtually any kind of interaction model. Nonetheless, the

performance of our interaction model is over 4 points better than random scores, indicating that our proposed scores impose an inductive bias which is well-suited for textual entailment.

For single-sentence encoders, our interaction model performs slightly worse than the MLP-based model. The fact that our models' performance is close indicates that our model is capturing nearly the same amount of information about NLI, and it has the advantage that all this learned information is encoded in the sentence embeddings, not in the MLP. When we move to inter-sentence encoders (+attn), the difference reduces in every case, and is virtually eliminated for SNLI. Since an inter-sentence encoder is able to model some interaction in the encoder itself, this suggests that the performance difference is in part due to the reduced power of our parameter-free interaction model compared to an MLP. For example, our interaction model with a single-sentence encoder does not seem powerful enough to model known artefacts in the datasets (Gururangan et al., 2018), which we will discuss further in Section 4.

### 3.4 Ablation Study

To better understand the effect of each of the scores, we performed an ablation study by evaluating models with subsets of our proposed scores, as shown in Table 2. The two similarity scores alone (S) perform rather poorly, indicating that similarity information alone is not enough to model entailment. In contrast, using only the three entailment scores (E,C,N) achieves very good performance, yielding within 0.66 points of the full model on MNLI and within 0.33 points of the full model on SNLI. Removing the neutral score had a surprisingly large impact on SNLI, but otherwise the neutral score is less important than the contradiction score, which is less important than the entailment score.

To further understand the roles of the different scores, consider the trained weights of the final classification layer for the model with only the entailment, contradiction and neutral scores (E,C,N):

$$
\boldsymbol{W}_c = \begin{array}{c} \\ E \\ N \\ C \end{array} \begin{array}{c} S_E \quad S_N \quad S_C \\ \begin{pmatrix} \boxed{+41.3} & +0.2 & -24.0 \\ -10.8 & -3.3 & -35.0 \\ -29.5 & +4.1 & \boxed{+60.0} \end{pmatrix} \end{array}, \quad \boldsymbol{b}_c = \begin{pmatrix} -26.4 \\ +21.0 \\ +5.3 \end{pmatrix}
$$

where the rows denote the labels and the columns denote the scores. The very large weights in the first and last columns indicate that indeed the entailment score predicts entailment and the contradiction score predicts contradiction. The neutral score weights interact with the biases to compensate for when the entailment and contradiction scores are both high.

## 4 Evaluation of Transfer Performance

One motivation for using sentence embeddings which are directly interpretable in terms of entailment and contradiction is that the resulting interaction model imposes a stronger inductive bias towards learning entailment and contradiction. Such inductive biases are often unhelpful when evaluating within the domain of a given dataset, particularly because the large NLI datasets, used in the previous section, contain annotation artefacts which allow high performance to be achieved without actually modelling entailment and contradiction (Gururangan et al., 2018). To evaluate the usefulness of this inductive bias for learning the more general regularities behind textual entailment, we evaluate our proposed model on its ability to generalise to a large number of different NLI datasets which do not share these same annotation biases.

We conduct these transfer experiments using sentence encoders which have been pretrained on very large amounts of unannotated text using a language modelling objective, namely the BERT model (Devlin et al., 2019). This pretraining has previously been demonstrated to help generalisation in a large number of tasks (Devlin et al., 2019), including state of the art results in NLI. In this section, we first evaluate our approach to learning sentence embeddings from NLI data by fine-tuning a pretrained BERT model, replicating the results from the previous section on the performance of the resulting NLI models[1]. We then evaluate the resulting sentence embeddings on their model's ability to generalise to a large number of other related tasks. In addition, we evaluate the resulting sentence embeddings on transfer to sentence classification tasks.

---

[1]We do not replicate the full range of experiments in this context because of the high computational cost of training with BERT pretrained models.

## 4.1 MODELS

The original BERT model of NLI (Devlin et al., 2019) encodes the premise and hypothesis together and uses the embedding of the CLS token at the last layer as input to a classifier. But this model does not produce individual sentence embeddings for the premise and hypothesis. To get a sentence embedding model based on BERT, we use the Sentence-BERT (SBERT) model proposed by Reimers & Gurevych (2019). Reimers & Gurevych (2019) encode each sentence individually using a pretrained BERT model, then obtain the final fixed-sized embedding by average-pooling over the outputs at the last layer. Average pooling was found to be better than max pooling or using the CLS token embedding.

For the baseline model (HM+SBERT), after obtaining an embedding for the premise and the hypothesis, we proceed similarly to Section 2.2. First the baseline computes heuristic matching features from the sentence embeddings, but instead of using an MLP classifier, it uses a log-linear classifier in accordance with Reimers & Gurevych (2019)[2]. Our model (Ours+SBERT) computes the 5 scores defined above and uses them in a log-linear classifier.

## 4.2 EXPERIMENTAL SETUP

We use BERT-Large as the foundation and finetune it with AdamW (Loshchilov & Hutter, 2019). We generally use a universal initial learning rate of 2e-5 for all heuristic matching models. When training our entailment-based model, a significantly larger learning rate of 0.006 was necessary for the parameters of the classifier. On SNLI, we found a learning rate of 2e-5 for training the BERT parameters and not initialising the output layer as described in Section 2.3 to work best. On MNLI, a learning rate of 2e-6 with initialisation worked best. In all cases, there is a warm-up phase lasting for 30% of the training dataset. For SNLI, we used a batch size of 8 whereas on MNLI we had to reduce the batch size to 4 because of limited memory.

## 4.3 NLI RESULTS

When training and testing on the same NLI corpora, finetuning a pretrained BERT model gives us the same pattern of results as in Section 3.3, as shown in Table 3. We see a small decrease in accuracies for both SNLI and MNLI for our much simpler interaction model, exacerbated by the fact that there is no interaction in the encoding stage. We hypothesise that this decrease is largely due to the inability of our simpler model to capture the annotation biases in these datasets. In the next section, we investigate this issue by testing on datasets which do not share the same annotation biases.

Table 3: Validation accuracies(for MNLI, on matched/mismatched sets), and number of parameters in the encoder (*#enc*) and MLP and/or classifier (*#mlp*).

| Model | #enc | #mlp | SNLI | MNLI |
|---|---|---|---|---|
| HM+SBERT | 13.8m | 12.3k | 87.2 | 76.1 / 76.2 |
| Ours+SBERT | 13.8m | 18 | 85.1 | 74.6 / 73.7 |
| **Test set results** | | | | |
| HM+SBERT | 13.8m | 12.3k | 86.0 | 75.2 / 75.6 |
| Ours+SBERT | 13.8m | 18 | 85.1 | 73.8 / 74.2 |

## 4.4 TRANSFER PERFORMANCE

Our model introduces a strong inductive bias for textual entailment that forces almost all the information about entailment to be contained in the sentence embeddings rather than the classifier. This is potentially very useful for transfer learning in two ways: transfer of the whole model to

---

[2]We found that using an MLP instead of a log-linear classifier resulted in the same validation performance on SNLI and worse validation performance on MNLI, particularly for the mismatched set.

out-of-domain NLI datasets, and transfer of the sentence embeddings as features for downstream tasks.

### 4.4.1 RESULTS ON OUT-OF-DOMAIN NLI DATASETS

To evaluate how well the baseline and proposed models generalise to solving textual entailment in domains which do not share the same annotation biases as the large NLI training sets used above, we take trained NLI models and test them on a number of different NLI datasets.

**Datasets:** We consider a total of 11 different NLI datasets. We use the 10 datasets studied by Poliak et al. (2018b). These datasets include MNLI, SNLI, SciTail (Khot et al., 2018), AddOneRTE (Pavlick & Callison-Burch, 2016), Johns Hopkins Ordinal Commonsense Inference (JOCI) (Zhang et al., 2017), Multiple Premise Entailment (MPE) (Lai et al., 2017), Sentences Involving Compositional Knowledge (SICK) (Marelli et al., 2014), and three datasets from White et al. (2017) which are automatically generated from existing datasets for other NLP tasks including: Semantic Proto-Roles (SPR) (Reisinger et al., 2015), Definite Pronoun Resolution (DPR) (Rahman & Ng, 2012), and FrameNet Plus (FN+) (Pavlick et al., 2015). We additionally consider the Recognising Textual Entailment dev dataset (RTE) from GLUE benchmark Wang et al. (2019), and the Quora Question Pairs (QQP) dataset[3], where the task is to determine whether two given questions are semantically matching (duplicate) or not. As in Gong et al. (2017), we interpret duplicate question pairs as an entailment relation and neutral otherwise. We use the same split ratio mentioned by Wang et al. (2017).

Since the datasets considered have different label spaces, when evaluating on each target dataset, we map the model's labels to the corresponding target dataset's space. We train all models on MNLI and evaluate their performance on other target datasets. MNLI contains three labels, contradiction, neutral, and entailment. Some of the datasets we consider contain only two labels. In the case of labels *entailed* and *not-entailed*, as in DPR, we map contradiction and neutral to the not-entailed class. In the case of labels *entailment* and *neutral*, as in SciTail, we map contradiction to neutral.

Table 4: Accuracy results of models transferring to new datasets. All models are trained on MNLI and tested on target test sets. $\Delta$ are absolute differences between our method and the baseline.

| Target Test Dataset | Methods | | |
| --- | --- | --- | --- |
| | Baseline | Ours | $\Delta$ Ours |
| RTE | 48.38 | 64.98 | +16.6 |
| JOCI | 41.14 | 45.58 | +4.44 |
| SCITAIL | 68.02 | 71.59 | +3.57 |
| SPR | 50.84 | 53.74 | +2.9 |
| QQP | 68.8 | 69.7 | +0.9 |
| DPR | 49.95 | 49.95 | 0 |
| FN+ | 43.04 | 42.81 | -0.23 |
| SICK | 56.57 | 54.03 | -2.54 |
| MPE | 48.1 | 41.0 | -7.10 |
| ADD-ONE-RTE | 29.2 | 17.05 | -12.15 |
| SNLI | 64.96 | 54.14 | -10.82 |

**Results** Table 4 shows the transfer performance of the baseline and proposed SBERT models to other NLI datasets. As shown in prior work (Belinkov et al., 2019a), the SNLI dataset has very similar annotation biases to the MNLI data which the models were trained on, so we do not expect any improvement in the relative performance of the two models for SNLI. Also, for the ADD-ONE-RTE dataset, both models essentially fail to generalise to this data, so the comparison is not very meaningful. For the remaining datasets, our proposed model performs better than the baseline on 5 datasets, the same on one, and worse on 3. On average, this represents a substantial improvement

---

[3]https://data.quora.com/First-Quora-Dataset-Release-QuestionPairs

in generalisation by using the proposed inductive bias, especially given that our model started out from a lower accuracy on the in-domain data.

### 4.4.2 RESULTS ON DOWNSTREAM TASKS

Because our model does not use a powerful MLP classifier on top of the sentence embeddings, it has to encode almost all the semantics about entailment in the sentence embeddings themselves. This should make the sentence embeddings more useful when used as features in downstream tasks.

To test this hypothesis, we evaluate the same SBERT-based sentence encoders from Section 4.4.1 on downstream tasks from the SentEval framework (Conneau & Kiela, 2018), which has previously been used to evaluate sentence embeddings trained on the textual entailment task (Conneau et al., 2017; Reimers & Gurevych, 2019). We use the default evaluation configuration, where for the single-sentence classification tasks a logistic regression classifier is trained on top of the given sentence embedding. We exclude the supervised sentence-pair tasks (STS-B, SICK-E, SICK-R, and MRPC) from our analysis because the SentEval implementation uses heuristic matching features for these experiments and thus is incompatible with our model. For related tasks, see the NLI experiments in section 4.4.1.

We report the downstream performance on supervised tasks in Table 5, and on unsupervised tasks in Table 6. In 10 out of 12 datasets, the sentence embeddings from our model achieve a substantial improvement over the sentence embeddings from the baseline. In the remaining two cases (TREC and STS15), the performance is approximately the same. These results support the hypothesis that our model produces sentence embeddings that carry more semantic information that are useful features in downstream tasks.

Table 5: Scores on supervised downstream tasks from SentEval toolkit attained by our model and the baseline.

| Model | MR | CR | MPQA | SUBJ | SST2 | SST5 | TREC |
|---|---|---|---|---|---|---|---|
| Ours+SBERT | **84.76** | **90.57** | **89.88** | **93.57** | **90.50** | **49.14** | 82.6 |
| HM+SBERT | 80.27 | 88.77 | 88.07 | 90.74 | 86.44 | 46.56 | **83.0** |
| $\Delta$ | +4.49 | +1.8 | +1.81 | +2.83 | +4.06 | +2.58 | -0.4 |

Table 6: Scores on unsupervised downstream tasks from SentEval toolkit attained by our model and the baseline. Numbers reported are Pearson correlations $\times 100$.

| Model | STS12 | STS13 | STS14 | STS15 | STS16 |
|---|---|---|---|---|---|
| Ours+SBERT | **61.25** | **60.58** | **66.18** | **66.85** | **67.40** |
| HM+SBERT | 53.39 | 50.65 | 62.89 | 66.53 | 63.51 |
| $\Delta$ | +7.86 | +9.93 | +3.29 | +0.32 | +3.89 |

To get a better understanding of what type of features our sentence embeddings encode, we evaluate them on the 10 linguistic probing tasks introduced by Conneau et al. (2018a). These tasks are grouped into three categories. The first two are 1) whether the embedding encodes *surface* information like word content (WC) or sentence length (SentLen), and 2) whether it encodes *syntactic* information like word order (BShift) or information about the parse tree structure (TopConst, Depth). The last category tests for *semantic* information. However, Conneau et al. (2018a) note that three of these tasks can arguably be solved mainly by considering surface information at the subword level (which are available to BERT), namely identifying the tense (Tense), and identifying the number of the subject (SubjNum) and object (ObjNum). The last two tasks require a solid semantic understanding of the encoded sentences, namely whether a random noun was replaced (odd-man-out (OMO)) and whether the coordinate clause was swapped with the main clause (CoordInv).

The results of our model in comparison with the baseline are shown in Table 7. With the exception of BShift, the baseline encodes substantially more surface and syntactic information. In contrast, our

model is superior at the semantically challenging tasks OMO and CoordInv, further supporting the hypothesis that our sentence embeddings encode more useful semantic features than the baseline.

Table 7: Scores on the probing tasks attained by our model and the baseline.

| Model | Length | WC | Depth | TopConst | BShift | Tense | SubjNum | ObjNum | OMO | CoordInv |
|---|---|---|---|---|---|---|---|---|---|---|
| Ours+SBERT | 54.94 | 46.60 | 28.27 | 55.66 | 79.58 | 82.56 | 74.16 | 73.89 | 63.33 | 66.42 |
| HM+SBERT | 75.19 | 55.14 | 33.86 | 65.76 | 70.78 | 83.67 | 78.70 | 78.43 | 55.72 | 62.50 |
| $\Delta$ | -20.25 | -8.54 | -5.59 | -10.1 | +8.8 | -1.11 | -4.54 | -4.54 | +7.61 | +3.92 |

## 5 CONCLUSION

Both the ablation study and the full NLI results demonstrate that our proposed entailment and contradiction scores are effective for modelling textual entailment. This parameter-free model of the relationship between premise and hypothesis forces training to put all information about entailment and contradiction in the sentence embeddings, rather than in the parameters of an MLP classifier, and gives those sentence embeddings a direct entailment-based interpretation. We also show that the inductive bias given by forcing the learned sentence embeddings to have this entailment-based interpretation results in better generalisations to other NLI datasets and more useful sentence embeddings for other semantic tasks.

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
