# OpenReview forum: "Learning Entailment-Based Sentence Embeddings from Natural Language Inference"
_ICLR.cc/2020/Conference — Reject_

### Official Review · AnonReviewer3 · 2019-10-20
**Official Blind Review #3**

**Rating:** 6

**Review:**

The paper proposes a few heuristic scorers to model entailment and contradiction, based on encoded sentence embeddings.

These scores include an entailment score, a contradiction score, a neutral score, and two similarity scores. They are defined heuristically, e.g., entailment score = geometric avg of such thing: 1 - sigma(premise not satisfied) * sigma(hypothesis satisfied). This is similar to fuzzy logic (for example, https://en.wikipedia.org/wiki/Fuzzy_logic) and some citations are needed in this regard.

Different from fuzzy logic, this paper learns whether an anonymous feature is true or false by NN encoders end-to-end. Thus, the model actually has enough power to extract those features suitable for fuzzy logic-like heuristic matching.

The experiments are well designed. I especially appreciate the comparison to random matching heuristics, which already exhibits non-trivial performance. This is very reasonable because the neural network underlying random matching heuristics is still learnable. However, the proposed matching heuristics achieve 7% improvement compared with random ones, showing the effectiveness of the approach.

The authors also have ablation test and experiments on out-of-domain datasets.

I have two concerns:

1. One limitation of this paper is that the heuristic matching scorers are pretty ad hoc to the inference task. The two similarity scores are not too novel, for example, sim_diff is the L1-distance between two vectors. Entailment, contradiction, and neutral scores are interesting, but hardly generalize to other sentence matching tasks (e.g., various IR applications).

2. I have a feeling that the importance of NLI is over-estimated. While logical reasoning is important in AI, NLI datasets are somehow degenerated, and existing solutions are basically connecting neural edges. As mentioned in the paper, NLI models do not transfer well to out-of-domain NLI samples, not to mention non-NLI tasks. It would be interesting to see if the well-designed heuristic matching scores could ease the underlying model, so that it learns more generic sentence embeddings in general.


Minor:

In references: Williams, Nagnia, Bowman: duplicate entry


**Experience Assessment:**

I have published in this field for several years.

**Review Assessment: Checking Correctness Of Derivations And Theory:**

I carefully checked the derivations and theory.

**Review Assessment: Checking Correctness Of Experiments:**

I carefully checked the experiments.

**Review Assessment: Thoroughness In Paper Reading:**

I read the paper thoroughly.

---

> ### Author Response · Authors · 2019-11-13
> **The fundamental nature of textual entailment**
>
> Thank you for your helpful comments and suggestions.
>
> We thank the reviewer for making the connection to fuzzy logic.  We are looking at this connection, in particular to understand the role of unknown values in fuzzy logic.  We will include a discussion in a later version of the paper.
>
> Regarding concern 1, that the scores are ad hoc to the inference task, and "Entailment, contradiction, and neutral scores are interesting, but hardly generalize to other sentence matching tasks (e.g., various IR applications)":
> We agree, in that the scores are designed to be specific to NLI, but we disagree that NLI is just another semantic task.  As we discuss in Section 1.1, entailment, and thus NLI, is fundamental to models of natural language semantics.  There is now a mini-industry of converting lots of different semantic tasks into NLI tasks (e.g. (Poliak et al., 2018a)).  Saying that the entailment score is ad hoc because it only measure information inclusion is like saying that the dot product is ad hoc because it only measures similarity.  More specifically for IR, it is easy to imagine a model of IR which says that a document is relevant if it entails the query (i.e. the information in the query is included in the information in the document).
>
> Regarding concern 2, that the importance of NLI is over-estimated, that NLI datasets are degenerate, and that NLI models do not transfer well:
> We share the view that existing NLI datasets are only partially representative of the fundamental problem of textual entailment.  That is why we focus in this paper on learning representations with a clear interpretation in terms of the fundamental task, rather than just maximising performance on the existing datasets, and why we evaluate the resulting inductive bias on transfer performance to datasets which are "degenerate" in different ways (i.e. have different biases).  In addition, our new transfer results with SentEval (discussed in our reply to Reviewer 1 and now reported in Section 4.4.2) demonstrate improved transfer performance on several different tasks, including semantic similarity (STS).

---

### Official Review · AnonReviewer2 · 2019-10-22
**Official Blind Review #2**

**Rating:** 6

**Review:**


***Update***
I'd like to thank the authors for responding to my questions and for the additional experiments. I think the new sentence embedding experiments make the paper quite a bit stronger - it would be interesting to scale them up to using SNLI + MNLI to see how much further they can go (right now they are still below Sentence-BERT which also was trained on SNLI in addition to MNLI). I think this paper is an interesting idea, and my only other main concern is the transfer results to other NLI datasets. I think it would be a good idea to confirm that the difference between the two approaches is due to biases perhaps though an error analysis. I am borderline on this paper, but I feel enough improvement has been made to raise my rating.


This paper proposes a new way to train sentence embedding models using NLI data such that very few parameters are used for classification. This is in contrast to prior work where an MLP is used. They define an interaction layer where operations are applied to the sentence embeddings to produce just 5 scores, which are then fed into a softmax layer for the final prediction.

The reason for their approach is they hypothesize that the the classification layers are encoding some of the information, presumably lessening the amount of information in the sentence embeddings and also preventing them from having a direct interpretation.

Their model does surprisingly well on the entailment datasets, only about 1-1.5 points or so worse than using a MLP, and so they indeed demonstrate that the sentence embeddings contain a lot of entailment information. However, I do have some concerns about their claims that this helps in transfer learning to other NLI tasks. Their main results show transfer performance comparing their approach and using an MLP, but it seems that overall, on all datasets, their approach transfers more poorly. Two datasets that they do worse on they try to discount for either having terrible, below random performance (which is true) and the other for having the same biases as MNLI. However, if that was the case, I don't see why their model would perform worse since both theirs and the baseline would benefit from having these biases, but their model performs about 11 points less.

So therefore, I don't find the transfer experiments convincing, though it is interesting how different the models do on some of these tasks - model performance is surprisingly task dependent

What I propose is for the authors to investigate if their sentence embeddings are in fact noticeably different than the ones trained in the more conventional matter. They could evaluate on sentence embedding and probing tasks (like SentEval) and see how the two models compare. It would be interesting to see wha encoded information differs between the models.

Im summary, I think this is an interesting experiment and it's nice to see that the MLP isn't doing a lot of heavy lifting (which also might be slightly counter to their hypothesis about the MLP containing a lot of entailment information). However, I find the transfer experiments unconvincing and the paper is short on analysis about when their model does better on transfer and when having an MLP helps, or how the learned sentence embeddings of the two models differ.

**Experience Assessment:**

I have published in this field for several years.

**Review Assessment: Checking Correctness Of Derivations And Theory:**

I carefully checked the derivations and theory.

**Review Assessment: Checking Correctness Of Experiments:**

I carefully checked the experiments.

**Review Assessment: Thoroughness In Paper Reading:**

I read the paper thoroughly.

---

> ### Author Response · Authors · 2019-11-13
> **New results with SentEval and inductive bias against learning annotation bias**
>
> Thanks for your helpful comments and suggestions.
>
> Regarding the comments "I do have some concerns about their claims that this helps in transfer learning to other NLI tasks" and "I don't see why their model would perform worse since both theirs and the baseline would benefit from having these biases":
> Our claim in the transfer experiments is that our model has an inductive bias which encourages it to learn entailment and contradiction and discourages it from learning the arbitrary functions needed to capture annotation biases.  The MLP, in contrast, can learn anything.  Thus, we expect the baseline to perform better when knowing the annotation biases of the training set is useful in the test set (SNLI), but we expect our model to perform better when it is more useful to know the true underlying NLI task which all these datasets have in common.  This is the nature of inductive biases.  Clearly we need to improve our presentation of this contribution.
>
> Regarding the comment "They could evaluate on sentence embedding and probing tasks (like SentEval)":
> We have now done this evaluation with SentEval, and have added these results in a new version of the paper in Section 4.4.2.  There are three types of experiments in SentEval, probing tasks, supervised transfer tasks and unsupervised transfer tasks.  Results for the probing tasks show that our embeddings are worse for recovering surface and syntactic characteristics, but are better for the semantic probing tasks, as desired.  For all the unsupervised transfer tasks and six out of seven of the supervised transfer tasks, our entailment-based sentence embeddings perform better than the baseline.  We do not include the sentence-pair tasks in these experiments because the SentEval implementation uses heuristic matching features for these experiments and thus is incompatible with our model.
>
> Regarding the comment "it's nice to see that the MLP isn't doing a lot of heavy lifting":
> We would simply like to clarify that the MLP classifier in the baseline is doing a lot of the work, but not in our model (where we do not have one).
>
> We hope that with these additional explanations and evaluations the reviewer will find our transfer experiments and analysis more convincing.

---

### Official Review · AnonReviewer1 · 2019-10-23
**Official Blind Review #1**

**Rating:** 6

**Review:**

This paper proposes an interesting approach towards learning NLI via parameter free operations over pairs of sentence embeddings. The authors propose entailment and contradiction operators that learn entailment and contradiction scores while training the parameters of the sentence encoders.

This is an interesting approach, however the experiments make me doubt the effectiveness of the proposed method. Admittedly, the authors do point out that at the cost of fewer training parameters, the proposed approach attains the same performance as NLI encoders with MLP based or attention based classifiers. However, the question becomes, how many fewer parameters are being learned to accept a performance that is in the same ball park but that does not exceed the SOTA.

The authors should have tried to an ablation type of approach in equation 1, to check if concatenation alone, element wise dot product alone or absolute difference alone or a combination of any two would work better with the scoring function.

This paper while taking a step in the right direction, seems a little premature for publication. That being said, the reported results my be of some value after all. It is hard to narrow down on the exact contributions of this paper.

**Experience Assessment:**

I have published in this field for several years.

**Review Assessment: Checking Correctness Of Derivations And Theory:**

I assessed the sensibility of the derivations and theory.

**Review Assessment: Checking Correctness Of Experiments:**

I carefully checked the experiments.

**Review Assessment: Thoroughness In Paper Reading:**

N/A

---

> ### Author Response · Authors · 2019-11-13
> **Clarifications on the main contributions and the numbers of parameters for various baselines**
>
> Thank you for your helpful comments and suggestions.
>
> Regarding the comments "the question becomes, how many fewer parameters" and "The authors should have tried to an ablation type of approach in equation 1":
> In Table 1 we report how many parameters are included in the interaction model and classifier ("#mlp"), and results for concatenation ("p,h") instead of equation 1 ("HM").  For both HM and concatenation, our interaction model and classifier has about five orders of magnitude fewer parameters.  And our model has much better accuracy than concatenation.  We don't know if there is a different subset of HM features which would perform as well as our model, but we know from previous work that this set of HM features performs better than such alternatives.  And it is clear that the number of parameters in any such classifier will still be over 4 orders of magnitude larger.  In contract, the results for the ablation of our model given in Table 2 show similar levels of accuracy for 18 (E,C,N,S), 12 (E,C,N) and 9 (E,C) parameters.  It is clear from these results that for any curve of accuracy versus parameters our model will be far superior.
>
> Regarding the comment "It is hard to narrow down on the exact contributions of this paper":
> As we tried to make clear in the last paragraph of Section 1, we believe that our main contributions are inducing sentence representations that are interpretable in terms of entailment (information inclusion) and contradiction, and providing an inductive bias which improves learning of the true NLI task at the expense of learning the annotation artefacts of individual NLI corpora.  The first claim is supported by the ablation study and classifier weight analysis in section 3.4.  The second claim is supported by the transfer performance reported in Section 4.4, which has now been extended with the results now in Section 4.4.2.  We will rewrite this summary.

---

### Decision · Program_Chairs · 2019-12-19

**Decision:**

Reject

**Comment:**

This paper proposes a method for learning sentence embeddings such that entailment and contradiction relationships between sentence pairs can be inferred by a simple parameter-free operation on the vectors for the two sentences.

Reviewers found the method and the results interesting, but in private discussion, couldn't reach a consensus on what (if any) substantial valuable contributions the paper had proven. The performance of the method isn't compellingly strong in absolute or relative terms, yielding doubts about the value of the method for entailment applications, and the reviewers didn't see a strong enough motivation for the line of work to justify publishing it as a tentative or exploratory effort at ICLR.

---

> ### Author Response · Authors · 2020-01-06
> **Meta-review of reviewing for Int. Conf. on Learning Representations**
>
> It is worrying for the field when reviewers for ICLR can't see any value in a novel and effective form of representation learning, and only consider engineering improvements.    This is a short-sited view of how science makes progress.
>
>     - James Henderson